# Efficient Quantization-Aware Training on Segment Anything Model in Medical Images and Its Deployment

Haisheng Lu[1][0000−0003−3978−1492], Yujie Fu[1][0009−0008−8597−7166], Fan Zhang[1][0000−0002−5032−6039], and Le Zhang[1][0000−0002−6930−8674]

University of Electronic Science and Technology of China, Chengdu, China
{luhaisheng, fuyujie}@std.uestc.edu.cn, {fan.zhang,lezhang}@uestc.edu.cn

**Abstract.** Medical image segmentation is a critical component of clinical practice, and the state-of-the-art MedSAM model has significantly advanced this field. Nevertheless, critiques highlight that MedSAM demands substantial computational resources during inference. To address this issue, the CVPR 2024 MedSAM on Laptop Challenge was established to find an optimal balance between accuracy and processing speed. In this paper, we introduce a quantization-aware training pipeline designed to efficiently quantize the Segment Anything Model for medical images and deploy it using the OpenVINO inference engine. This pipeline optimizes both training time and disk storage. Our experimental results confirm that this approach considerably enhances processing speed over the baseline, while still achieving an acceptable accuracy level. The training script, inference script, and quantized model are publicly accessible at https://github.com/AVC2-UESTC/QMedSAM.

**Keywords:** Quantization-Aware Training · Segment Anything Model.

## 1 Introduction

Drawing inspiration from the remarkable achievements of foundation models in natural language processing, researchers at Meta FAIR introduced a versatile foundation model for image segmentation, termed the Segment Anything Model (SAM) [3]. It is widely recognized that foundation models in any domain often confront challenges stemming from limited data diversity. Despite the considerable scale of the dataset utilized to train SAM (referred to as the SA-1B dataset), comprising over one billion masks, the model's performance fell short in medical image segmentation tasks [10]. This shortfall can be attributed in part to the composition of the SA-1B dataset, which primarily comprises photographs of natural scenes captured by cameras, thus lacking the nuanced features characteristic of medical images. In response to this challenge, Ma et al. curated a diverse and extensive medical image segmentation dataset encompassing 15 modalities, upon which they fine-tuned SAM [10]. Their refined model, dubbed MedSAM, represents a significant step forward in addressing this discrepancy. However,

despite its advancements, MedSAM still grapples with several unresolved challenges. For instance, the training dataset suffers from extreme modality imbalances, the model encounters difficulties in accurately segmenting vessel-like branching structures, and the practicality of text prompts remains limited.

The focus of the CVPR 2024 MedSAM on Laptop Challenge is on enhancing the inference speed of MedSAM. The Segment Anything Model comprises three core components: an image encoder responsible for transforming input images into image embeddings, a prompt encoder that converts prompts into prompt embeddings, and a mask decoder tasked with generating low-resolution masks from image embeddings and prompt embeddings. Notably, in the initial prototype of MedSAM, the image encoder is notably more resource-intensive than the other two components. Consequently, various alternative backbones have been proposed to replace the original image encoder, such as the ViT-Tiny architecture adopted by MobileSAM [15] and EfficientViT in EfficientViT-SAM [17]. The challenge's baseline model (LiteMedSAM) incorporates a distilled ViT-Tiny image encoder, albeit with slight adjustments compared to MobileSAM. A summary of the parameters of the different submodules is provided in Table 1.

**Table 1.** Parameters of different submodules in LiteMedSAM and MedSAM

| Parameters | Image Encoder | Prompt Encoder | Mask Decoder |
|---|---|---|---|
| LiteMedSAM | 5.7M | 6.2K | 4.1M |
| MedSAM | 89.7M | | |

In addition to optimizing the backbones of SAM, we pursued an alternative approach to expedite inference: quantization. Quantization offers several benefits, including reducing parameter sizes, increasing inference speed, and decreasing power consumption during inference. There are two primary paradigms for quantizing neural networks: post-training quantization (PTQ) [16] [8] [1] and quantization-aware training (QAT) [2] [13]. PTQ involves converting a pretrained floating-point model directly into a low-precision one by calibrating the model using a batch of calibration data. This method is generally faster since it does not require re-training, and the precision of the quantized model largely depends on the calibration process. On the other hand, QAT integrates quantization and de-quantization nodes into the computational graph, enabling the training of the model while preserving its accuracy after quantization. To ensure prediction accuracy, we chose QAT to quantize SAM.

The attention blocks of transformers serve as the principal components in the backbone of SAM. Several methods have been proposed to enhance the accuracy of quantized transformers. Li et al. introduced an information rectification module and a distribution-guided distillation scheme tailored for fully quantized vision transformers [5]. Liu et al. discovered that incorporating fixed uniform noise into the values being quantized can significantly mitigate quantization errors under provable conditions [6]. In this study, we have chosen to leverage the Xilinx Brevitas framework [11]. This framework offers an excellent workflow,

encompassing quantization-aware training through to development on inference engines.

The main contributions of this paper are listed as follows:

1. We propose a quantized LiteMedSAM model with comparable average accuracy, and alleviate the imbalance across different modalities.
2. An optimized online dataset is proposed to replace the offline baseline, yielding a significant reduction in disk storage requirement.
3. Experiments have been proposed to prove that a small subset of the training dataset can maintain the accuracy of the quantized model, making it more efficient in training.
4. The quantized model is deployed on the OpenVINO inference engine, enabling it to compete effectively with other models in the challenge.

## 2   Method

### 2.1   Preprocessing

The dataset comprises three types of medical images: grayscale images, RGB images, and 3D images. 3D images are split into individual 2D clips along the z-axis, with each clip treated as a grayscale image. To standardize the grayscale format with the RGB format, grayscale images are duplicated across the red, green, and blue channels. Subsequently, RGB images are resized, padded to dimensions of $256 \times 256$, and finally normalized. It's important to note that in the baseline approach, RGB images undergo normalization before padding with zeros. In this case, the padded value is equivalent to the minimum value of the image instead of zero.

We've implemented some optimizations in the dataloader to enhance efficiency during both training and inference. For the training process, in the baseline approach, all compressed 3D npz files are decompressed along the z-axis, which demands approximately 10TB of disk storage. This overhead is significantly disproportionate to the size of the original dataset, which is only around 160GB. To mitigate this inefficiency, we propose indexing each 3D clip along the z-axis and employing a binary search algorithm to locate the target 2D clip when necessary. By adopting this strategy, we distribute the decompression time across each batch of training data, resulting in substantial savings in disk storage. Additionally, considering that our machine typically processes one batch of data in approximately one second, the computational cost of decompression becomes negligible.

In terms of inference, the baseline method iterates through each 3D prompt box individually. However, when 3D boxes intersect along the z-axis, the baseline recalculates image features. Given that the image encoder constitutes the most computationally intensive aspect of SAM, we propose to preprocess all the 3D boxes into 2D boxes corresponding to 2D clips. This approach ensures that the image embedding of each 2D clip is computed only once, optimizing computational resources. In addition, the challenge has an 8GB limit on the Docker

running memory. Experiments show that LiteMedSAM will exceed the memory limit when the number of boxes approaches 100. Since the maximum number of boxes is 255, we propose a block partition algorithm along the batch axis of boxes. This algorithm allows users to specify the maximum running batch size to prevent exceeding the memory limit.

## 2.2   Proposed method

We propose to quantize the baseline model LiteMedSAM using QAT. While neural networks consist of various components beyond just matrix multiplications, it's within these operations that the peak of computational complexity resides. Therefore, nearly every QAT method focuses on quantizing inputs and weights during matrix multiplications, such as in linear layers, convolution layers, and attention blocks. In contrast, operations involving biases, activation layers, and normalization layers are typically performed per element. While the quantization of these layers can be selective, in our proposed quantized model, we opt to retain all these layers as floating-point, with only matrix multiplications in the image encoder and the mask decoder being quantized. The reason we choose not to quantize the prompt encoder lies in the fact that its parameter size is over 1000 times smaller than the other two modules, as indicated in Table 1. Some of the most common quantized sub-structures are illustrated in Figure 1.

Since quantization is non-differentiable, we employ the straight-through estimator (STE) methodology, as demonstrated in previous works [7]. In STE, incoming gradients are directly passed through a threshold operation to become outgoing gradients. For each quantization node, we propose an 8-bit symmetric per-tensor signed integer activations quantizer with a learned floating-point scale factor. This scale factor is initialized from runtime statistics.

## 2.3   Model Inference and Post-processing

Upon completion of quantization-aware training, Brevitas provides exceptional toolchains for exporting quantized models to diverse backends.

While the standard QuantizeLinear-DeQuantizeLinear (QCQ) representation for quantization in ONNX exists, Brevitas has extended this to QuantizeLinear-Clip-DeQuantizeLinear (QCDQ). With this extension, researchers can confine the range of quantized values. Therefore, we propose exporting the quantized LiteMedSAM to ONNX in the QCDQ representation.

While numerous inference engines support the ONNX format, not all of them are compatible with QCDQ. Given that the challenge mandates CPU inference, we narrow down the options to ONNX Runtime and OpenVINO. An experiment on inference speed between these two inference engines is detailed in Section 4.1. Based on the results, we ultimately opt for OpenVINO. Model caching is also supported by OpenVINO. This strategy can reduce the resulting delays at application startup, making it considerably suitable for accelerating in this challenge [4] [12].

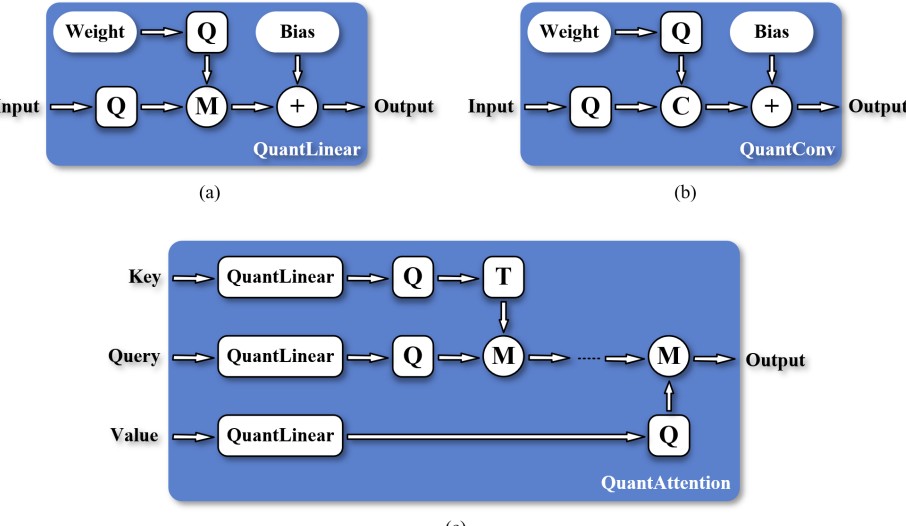

**Fig. 1.** Common quantized sub-layers. (a) quantized linear layer; (b) quantized convolutional layer; (c) quantized attention block. Circles in the figure represent corresponding calculations: M stands for matrix multiplication, C stands for convolution, and T stands for transpose. Operations involving quantization are represented by round rectangles in the figure. The inputs and output of all the sub-layers depicted in the figure are floating-point tensors.

SAM generates a $256 \times 256$ mask for the provided image and prompt. We binarize the floating-point values to either 0 or 1, crop the padding, and subsequently resize the low-resolution mask to the original dimensions of the input image.

## 3    Experiments

### 3.1    Dataset and sampler

We employed the challenge dataset for training, while the evaluation dataset was obtained by partitioning it at a ratio of one-tenth. The dataset comprises 11 modalities, and their sizes (prior to partitioning into training and evaluation datasets) are summarized in Table 2. An evident issue arises from the significant imbalance in sample numbers across modalities.To address this imbalance and prevent bias or overfitting in the quantized model, as well as to expedite training, we propose randomly sampling $N_s$ 2D clips from each modality in each epoch. Additionally, these samples undergo random horizontal and vertical flips for data augmentation.

**Table 2.** Samples of modalities in the training dataset (including the additional datasets released in the post-challenge task). 3D modalities are counted with the number of 2D clips on the z-axis.

| 3D Modalities | CT | MR | PET | |
|---|---|---|---|---|
| Samples | 1218411 | 236804 | 89059 | |
| 2D Modalities | Endoscopy | X-Ray | Dermoscopy | US |
| Samples | 43443 | 34893 | 3694 | 1646 |
| 2D Modalities | OCT | Mammography | Fundus | Microscopy |
| Samples | 1436 | 1233 | 1057 | 1000 |

### 3.2    Metrics and loss functions

The accuracy of the model is evaluated using the Dice Similarity Coefficient (DSC) and the Normalized Surface Distance (NSD), while efficiency is measured through running time analysis. These metrics are collectively utilized to compute the ranking. In the training phase, we mainly employ a combination of the Dice loss and focal loss. This decision is based on the robustness demonstrated by compound loss functions in various medical image segmentation tasks, as evidenced in prior research [9].

### 3.3    Training protocols

The training procedure includes three stages.

In stage one, our goal is to train the quantized image encoder while keeping the floating-point prompt encoder and the mask decoder frozen. Apart from the loss function mentioned in section 3.2, we further distill the image encoder from MedSAM and introduce the distillation loss. This loss is calculated as the product of the mean squared error and the intersection over union ratio across the image embeddings generated from the teacher and student models.

In stage two, we propose to train the quantized mask decoder by concatenating it with the best-trained quantized image encoder from stage one and the floating-point prompt encoder.

In the final stage, the whole model undergoes an end-to-end fine-tuning for further fitting with the dataset.

For each stage, we propose employing linear learning rate warm-up for $N_w$ epochs, commencing at 1% of the initial learning rate. Additional training details are summarized in Table 3. This warm-up period is followed by a cosine annealing scheduler for $N_a$ epochs. The minimum learning rate of the cosine annealing scheduler is set to 0.1% of the initial learning rate, and the half-period of the cosine function is determined as $N_a - 1$. Once the quantization-aware training process is completed, we evaluate the checkpoint of each epoch on the evaluation dataset and select the best-performing one. Additional training details are summarized in Table 3.

**Table 3.** Training protocols. Values separated by vertical bars in the table correspond to stages 1 ∼ 3.

| | |
|---|---|
| Pre-trained Model | LiteMedSAM (the baseline) |
| Batch size | 2 \| 4 \| 2 |
| DDP world size | 4 |
| Samples of each modality ($N_s$) | 900 |
| Optimizer | SGD (momentum=0.9) |
| Total epochs | 14 |
| Initial learning rate | 0.01 |
| Warm-up epochs ($N_w$) | 5 |
| Cosine annealing epochs ($N_a$) | 10 |
| Training time | 5 \| 2.5 \| 1 hours |

### 3.4   Environment settings

The development environments and requirements are presented in Table 4.

**Table 4.** Development environments and requirements.

| | |
|---|---|
| System | Ubuntu 20.04.3 LTS |
| CPU | Intel(R) Xeon(R) Gold 5218R CPU@2.10GHz |
| RAM | 16×32GB |
| GPU | 4×NVIDIA GeForce RTX 3090 |
| CUDA version | 12.2 |
| Programming language | Python 3.11 |
| Deep learning framework | PyTorch 2.0.1 |
| Specific dependencies | Brevitas 0.10.3 |
| Code | https://github.com/AVC2-UESTC/QMedSAM |

## 4    Results and discussion

### 4.1    Inference speeds of different engines

The challenge evaluates models on an Intel Xeon W-2133 CPU (6c12t@3.8GHz), while we use an Intel Core i7-8750H CPU (6c12t@4.1GHz) that offers comparable performance because we do not have an identical environment. We test each variant with a single image and a prompt box. The inference speeds of various methods are detailed in Table 5.

**Table 5.** Inference speed of different LiteMedSAM variants.

| Method | Inference time |
|---|---|
| LiteMedSAM inferenced on PyTorch | 1.180s |
| LiteMedSAM exported to ONNX and inferenced on ONNX Runtime | 0.787s |
| LiteMedSAM exported to ONNX and inferenced on OpenVINO | 0.574s |
| Quantized LiteMedSAM inferenced on ONNX Runtime | 0.769s |
| Quantized LiteMedSAM inferenced on OpenVINO | 0.585s |

The results indicate that the quantized model does not exhibit the fastest runtime. This is because that our hardware is not optimized for quantized operations, resulting in slower execution compared to standard floating-point operations. For comparison purposes, the inference speeds of both floating-point and quantized versions of MedSAM (which is substantially larger than LiteMed-SAM) are provided in Table 6. Interestingly, in this case the quantized model outperforms the floating-point model.

Given the comprehensive advantages of quantization, it is evident that deploying the quantized LiteMedSAM on the OpenVINO inference engine effectively addresses the requirement for medical image segmentation "on laptop".

### 4.2    Quantitative results on validation set

Table 7 presents the performance of the proposed three stages in comparison with the baseline model on the public validation dataset.

**Table 6.** Inference speed of different MedSAM variants.

| Method | Inference time |
|---|---|
| MedSAM inferenced on PyTorch | 10.181s |
| MedSAM exported to ONNX and inferenced on ONNX Runtime | 5.707s |
| MedSAM exported to ONNX and inferenced on OpenVINO | 4.202s |
| Quantized MedSAM inferenced on ONNX Runtime | 4.531s |
| Quantized MedSAM inferenced on OpenVINO | 3.558s |

On average, the quantized model scores comparably on DSC and slightly higher on NSD. We highlight the modalities with significant differences in their accuracy. In particular, the quantized model has degraded performance by around 3% and 5% in MR and US, but shows gains of approximately 10% and 9% improvement in PET and Microscope. It is evident that, to a certain extent, the proposed method has effectively addressed the performance imbalance of the baseline model across various modalities, which was caused by the dataset's inherent imbalance.

**Table 7.** Quantitative evaluation results on the validation dataset.

|  | Stage 3 | | Stage 2 | | Stage 1 | | Baseline | |
|---|---|---|---|---|---|---|---|---|
|  | DSC | NSD | DSC | NSD | DSC | NSD | DSC | NSD |
| CT | 89.35% | 92.84% | 89.73% | 93.23% | 89.86% | 93.27% | 90.78% | 93.08% |
| MR | 82.41% | 87.29% | 82.73% | 87.76% | 82.91% | 87.87% | **86.43%** | **90.37%** |
| PET | **64.80%** | **56.33%** | 63.37% | 49.52% | 63.86% | 48.75% | 57.64% | 43.05% |
| US | 87.87% | 92.41% | 87.93% | 92.50% | 87.88% | 92.39% | **94.54%** | **96.62%** |
| X-Ray | 78.73% | 84.19% | 78.14% | 83.80% | 78.62% | 84.31% | 79.15% | 84.46% |
| Dermoscopy | 91.71% | 93.31% | 92.15% | 93.75% | 92.12% | 93.70% | 91.59% | 93.21% |
| Endoscopy | 93.37% | 96.61% | 93.56% | 96.71% | 94.08% | 97.12% | 94.81% | 97.70% |
| Fundus | 93.24% | 94.66% | 93.85% | 95.19% | 92.97% | 94.30% | 94.40% | 95.77% |
| Microscope | **70.11%** | **77.35%** | 71.77% | 79.21% | 72.77% | 80.18% | 60.54% | 65.12% |
| Average | **83.51%** | **86.11%** | 83.69% | 85.74% | 83.90% | 85.76% | 83.32% | 84.38% |

A comparison of inference speeds for specific cases between the baseline and the proposed method is presented in Table 8. The results highlight a notable acceleration achieved by the quantization method.

### 4.3 Qualitative results on validation set

Two sets of successful segmentation results are depicted in Figure 2. It can be observed that the proposed quantized model performs better in matching the ROI than the floating-point counterpart. Figure 3 illustrates two sets of challenging cases. In these cases, the segmentation results of the proposed quantized model align more closely with the ground truth ROI compared to the baseline. However, since the baseline prediction results were significantly distant from the ground truth, the correction was unsuccessful.

**Table 8.** Quantitative efficiency in terms of inference running time (seconds). MLE stands for Memory Limit Exceeded.

| Case ID | Size | Objects | Baseline | Proposed |
|---|---|---|---|---|
| 3DBox_CT_0566 | (287, 512, 512) | 6 | 591.1 | 142.1 |
| 3DBox_CT_0888 | (237, 512, 512) | 6 | 168.7 | 51.0 |
| 3DBox_CT_0860 | (246, 512, 512) | 1 | 23.4 | 12.4 |
| 3DBox_MR_0621 | (115, 400, 400) | 6 | 245.6 | 51.5 |
| 3DBox_MR_0121 | (64, 290, 320) | 6 | 168.4 | 31.4 |
| 3DBox_MR_0179 | (84, 512, 512) | 1 | 22.5 | 11.9 |
| 3DBox_PET_0001 | (264, 200, 200) | 1 | 15.1 | 7.3 |
| 2DBox_US_0525 | (256, 256, 3) | 1 | 1.6 | 0.7 |
| 2DBox_X-Ray_0053 | (320, 640, 3) | 34 | 9.2 | 1.8 |
| 2DBox_Dermoscopy_0003 | (3024, 4032, 3) | 1 | 6.5 | 1.1 |
| 2DBox_Endoscopy_0086 | (480, 560, 3) | 1 | 2.3 | 0.6 |
| 2DBox_Fundus_0003 | (2048, 2048, 3) | 1 | 3.5 | 0.7 |
| 2DBox_Microscope_0008 | (1536, 2040, 3) | 19 | 15.6 | 1.6 |
| 2DBox_Microscope_0016 | (1920, 2560, 3) | 241 | MLE | 14.0 |

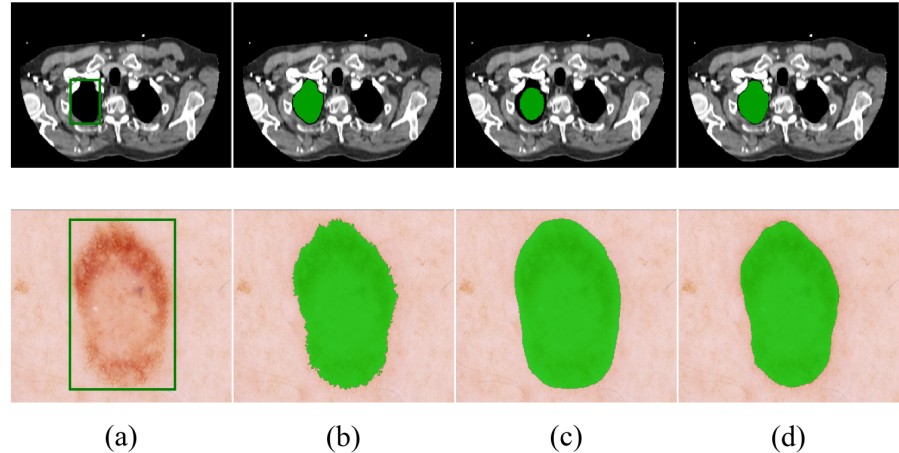

(a)                (b)                (c)                (d)

**Fig. 2.** Good segmentation results. (a) Image and box; (b) Ground truth; (c) Baseline; (d) Proposed method.

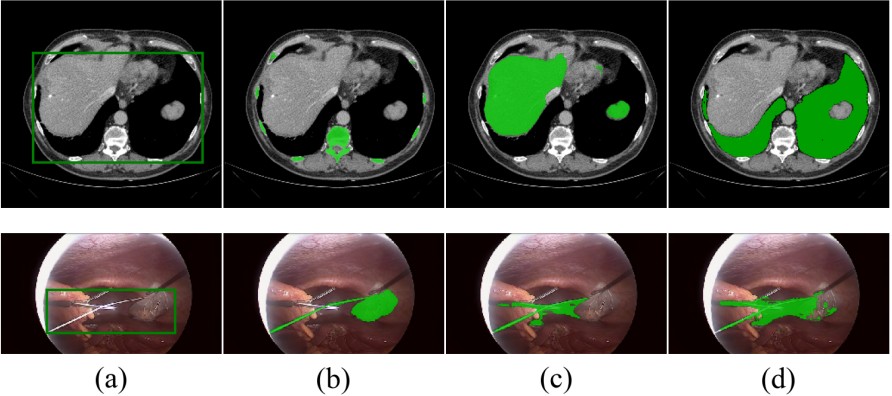

|  (a)  |  (b)  |  (c)  |  (d)  |

**Fig. 3.** Bad segmentation results. (a) Image and box; (b) Ground truth; (c) Baseline; (d) Proposed method.

### 4.4  Ablation Study

Training a Segment **Anything** Model from scratch requires a huge mass of data. However, the proposed quantization-aware training procedure starts with a pre-trained model. Reducing the number of samples $N_s$ from each modality, especially from the larger modalities, certainly benefits in saving training time. However, it still raises questions about its influence on the precision of the quantized model. In this section we propose an ablation study to explore the balance between efficiency and accuracy.

To describe the variation of samples from different modalities clearly, we will use $N_s(m)$ to represent the number of samples from modality $m$. The total samples of modality $m$ is denoted by $N_m(m)$, and the complete set of modalities is denoted by $M$. The strategy of the proposed method can be described as

$$N_s(m) = \min_{i \in M} N_m(i).$$

The ablation study introduces a strategy that enlarges $N_s(m)$ to one-tenth of $N_m(m)$, in particular,

$$N_s(m) = \max\left\{\frac{N_m(m)}{10}, \min_{i \in M} N_m(i)\right\}.$$

The metrics of the three stages in the ablation study are summarized in Table 9. Compared with Table 7 (we provide the average metrics of the proposed method in the last row of Table 9), the results indicate that increasing $N_s$ does not result in a significant improvement, underscoring the efficiency of the proposed QAT pipeline in terms of training time.

**Table 9.** Evaluation results of the ablation study on the validation dataset.

|  | Stage 3 | | Stage 2 | | Stage 1 | |
|---|---|---|---|---|---|---|
|  | DSC | NSD | DSC | NSD | DSC | NSD |
| CT | 88.71% | 92.37% | 87.02% | 91.14% | 88.81% | 92.45% |
| MR | 81.55% | 86.48% | 80.91% | 86.18% | 81.61% | 86.40% |
| PET | 64.41% | 55.09% | 64.35% | 54.73% | 65.21% | 52.62% |
| US | 86.93% | 91.76% | 86.09% | 90.87% | 87.43% | 91.85% |
| X-Ray | 79.07% | 84.53% | 76.44% | 82.13% | 76.18% | 81.83% |
| Dermoscopy | 91.65% | 93.24% | 92.63% | 94.20% | 91.75% | 93.34% |
| Endoscopy | 93.42% | 96.65% | 93.99% | 97.09% | 92.65% | 95.84% |
| Fundus | 93.18% | 94.59% | 96.05% | 97.20% | 92.93% | 94.33% |
| Microscope | 72.29% | 79.64% | 71.03% | 78.52% | 72.94% | 80.40% |
| Average | 83.47% | 86.04% | 83.17% | 85.79% | 83.28% | 85.45% |
| Proposed | 83.51% | 86.11% | 83.69% | 85.74% | 83.90% | 85.76% |

### 4.5   Results on final testing set

The testing results are summarized in Table 10. The proposed quantized model exhibits a marginal decrease but much more balance in the average accuracy. Additionally, the inference efficiency has been significantly optimized under the same backbone. Compared with Table 7, we can observe that the model's performance on different modalities varies between the validation set and the testing set. However, the trend of balance across modalities remains consistent.

**Table 10.** Evaluation results on the test dataset.

|  | Proposed | | | Baseline | | |
|---|---|---|---|---|---|---|
|  | DSC | NSD | RunTime | DSC | NSD | RunTime |
| CT | **69.74%** | **71.91%** | 11.78s | 55.75% | 58.48% | 38.78s |
| MR | **69.33%** | **61.77%** | 6.20s | 64.80% | 62.75% | 18.57s |
| X-Ray | 80.13% | 89.56% | 2.50s | **85.51%** | **94.40%** | 9.95s |
| Endoscopy | 89.81% | 93.15% | 2.18s | **94.41%** | **96.95%** | 7.56s |
| Fundus | 79.05% | 81.28% | 2.23s | **87.47%** | **89.58%** | 8.77s |
| Microscope | 79.68% | 81.72% | 2.58s | **84.36%** | **86.15%** | 16.34s |
| OCT | 72.72% | 79.50% | 2.24s | 73.31% | 80.20% | 8.39s |
| PET | 76.53% | 67.52% | 4.87s | 76.94% | 66.98% | 14.90s |
| US | **87.49%** | **92.09%** | 2.75s | 85.24% | 89.73% | 8.96s |
| Average | 78.28% | 79.83% | 4.15s | 78.64% | 80.58% | 14.69s |

### 4.6   Limitation and future work

Experimental results have shown a significant decrease in performance in certain modalities with larger amounts of data, and the accuracy of the least accurate modalities still lags far behind the average. Hence a more accurate and modality-balanced quantization is expected. On the other hand, the floating-point model

runs faster on the OpenVINO inference engine. We did explain a bit about this above, but beyond that, Brevitas also provides an excellent workflow to export the quantized model to FINN for dataflow acceleration on Xilinx FPGAs. Quantized models promise faster and more energy-efficient inference on a customized hardware platform.

## 5    Conclusion

In this paper, we present an efficient pipeline for quantizing LiteMedSAM and deploying it on the OpenVINO inference engine. Objective experiments have conclusively shown that our method significantly accelerates the baseline while maintaining an acceptable level of accuracy. Future endeavors will focus on enhancing the speed of the floating-point backbone, further alleviating the imbalance across different modalities, and deploying the quantized model on customized hardware platforms.

**Acknowledgements** We express our gratitude to all the data owners for making the medical images publicly available, and to CodaLab [14] for hosting the challenge platform.

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

**Table 11.** Checklist Table. Please fill out this checklist table in the answer column.

| Requirements | Answer |
|---|---|
| A meaningful title | Yes |
| The number of authors ($\leq$6) | 4 |
| Author affiliations and ORCID | Yes |
| Corresponding author email is presented | Yes |
| Validation scores are presented in the abstract | Yes |
| Introduction includes at least three parts: background, related work, and motivation | Yes |
| A pipeline/network figure is provided | Figure 1 |
| Pre-processing | Page 3 |
| Strategies to data augmentation | Page 6 |
| Strategies to improve model inference | Page 4 |
| Post-processing | Page 4 |
| Environment setting table is provided | Table 4 |
| Training protocol table is provided | Table 3 |
| Ablation study | Page 11 |
| Efficiency evaluation results are provided | Table 5 8 |
| Visualized segmentation example is provided | Figure 2 3 |
| Limitation and future work are presented | Yes |
| Reference format is consistent | Yes |
| Main text >= 8 pages (not include references and appendix) | Yes |