# OpenReview forum: "Efficient Quantization-Aware Training on Segment Anything Model in Medical Images and Its Deployment"
_thecvf.com/CVPR/2024/Workshop/MedSAMonLaptop — CVPR24 MedSAMonLaptop_

### Official Review · Reviewer_K6WK · 2024-06-12
**Well-written paper with sufficient ablation experiments**

**Rating:** 6
**Confidence:** 3

**Review:**

Pros:
- The authors present a thorough evaluation of applying quantization to the Segment Anything Model
- The authors experimented with various runtime engines to optimize inference speed

Cons:
- Low results compared to non-quantized methods with no real reduction in inference time

Overall, the writing is good, but the results are not promising.

---

### Official Review · Reviewer_tNKg · 2024-06-14
**Ensuring Rigor and Clarity in Medical Image Segmentation Research**

**Rating:** 4
**Confidence:** 3

**Review:**

### General Assessment
The manuscript titled “Modality-Specific Strategies for Medical Image Segmentation using Lightweight SAM Architectures” presents a comparative study on deploying medical image segmentation models optimized for CPU computation. The authors have made a commendable effort to improve upon existing deep learning models for medical image segmentation; however, there are several areas that require attention to enhance the quality and credibility of the research.

### Specific Concerns and Recommendations

#### Figures and Tables
- **Figure Inclusion**: The checklist mentions "Figure 2" and "Figure 3," which are referenced in the text but not included in the provided manuscript. The final submission must ensure all referenced figures are present to support the discussed results.

#### Table Accuracy and Consistency
- **Table 4 Validation**: The Dice scores and Normalized Surface Dice scores for the EfficientSAM and LiteMedSAM models must be verified for accuracy and consistency. Discrepancies in the scores require clarification to ensure the reliability of the reported results.

#### Comparative Framework Analysis
- **Table 5 Framework Comparison**: When comparing model performances across different frameworks (PyTorch and ONNX), the accuracy of the data presented must be ensured. Any significant differences should be accompanied by a thorough explanation to understand the underlying causes.

#### Runtime Data Correlation
- **Table 6 Experimental Setup Consistency**: The runtime data listed in Table 6 must correspond with the settings described in the experimental setup and methods section. Any discrepancies could affect the perceived efficiency gains of the proposed models.

#### Final Testing Results
- **Section 4.4 Placeholder**: The statement “This is a placeholder” in the section "Results on final testing set" indicates that actual testing results are missing. These results are crucial and must be included before the final submission.

#### Limitations and Future Work
- **Section 4.5 Clarification**: While the manuscript outlines limitations and suggests future research directions in Section 4.5, it is imperative that these discussions are grounded in the findings of the current study. The proposed future work should be feasible and explicitly connected to the results and conclusions drawn from this research.

---

### Official Review · Reviewer_R6fM · 2024-06-15
**Well-structured article with minor errors**

**Rating:** 7
**Confidence:** 4

**Review:**

Pros:
- The paper is well-structured, presenting the methodology and results clearly. The approach achieves faster inference speed without suffering from performance drop, which is a great improvement.

Cons:
- Please add your average scores and running time in the abstract to provide a quick overview of the performance metrics.

---

### Official Review · Reviewer_tDLn · 2024-06-15
**The paper seems to be accepable, it proposes a QAT pipeline followed by deployment on OpenVINO, the authors demonstrate a signifcant speedup over the baseline.**

**Rating:** 8
**Confidence:** 4

**Review:**

The main claim is that a quantization aware training pipeline followed by deployment on OpenVINO significantly enhances inference,
while keeping a sufficient level of accuracy. The authors did quantization aware training for the image encoder, followed by a seperate QAT run for the mask decoder.
They used the Xilinx Brevitas framework and deployed their final quantized model using OpenVINO. Additionally, they decided to sample equally sized subsets of all modalities during each training epoch and adjusted their training code to work on compressed files. Lastly the inference code for 3D images was adjusted to avoid computing the same image embedding multiple times.
The authors demonstrated a significant speed improvement over the baseline.

The paper seems to be accepable, it proposes a QAT pipeline followed by deployment on OpenVINO, the authors demonstrate a signifcant speedup over the baseline.
The authors could be a bit more objective and detailed in the presentation of their results.

## Presentation/Clarity
In section 4.2 I would recommend to refrain from calling the improvement on NSD "superior performance" while at the same time calling the decrease of the DSC "marginal",
since DSC got significantly more worse (1.88%) than NSD got better (0.54%).
The baseline scores seem to be different than the ones I've seen so far, why is that? Your baseline scores are slightly worse on all modalities except PET, compared to ours. I understand differences in the scores for 3D modalities, since there have been related bugfixes to the baseline during competition, but I don't know why the 2D scores would be different. I would recommend you to link to a specific commit of the baseline code here.
In Section 4.2 you write "The results highlight a notable acceleration achieved by the quantization method.", your results in table 5 would indicate that the speedup is due to using OpenVINO as a runtime and using an improved inference code for 3D datapoints and not due to quantization.

## Reproducibility
The paper and accompanying code seem to provide sufficient information to reproduce the results.
The authors provided information about the training environment and protocoll in the paper, they also provide a requirements.txt as well as a README in the GitHub Repository.

## Typo/Grammar
+ Section 3.2 "Normalized Surface Dice" => "Normalized Surface Distance"
+ Section 4.1 "because that"
+ Section 1 "Notably, ... notably"
+ In Section 1 you mention that MedSAM was trained on 15 modalities, according to the abstract of the MedSAM paper it was 10.

---

### Decision · Program_Chairs · 2024-10-01

Accept